# Gain-Phase Error-Calibrated Piezoelectric Sensor Array-Based Impact Localization on Stiffened Curved Composite Structures

**DOI:** 10.3390/s22155879

**Published:** 2022-08-05

**Authors:** Zhiling Wang, Jinyu Zhou, Yongteng Zhong, Chaoyue Li

**Affiliations:** 1School of Mechanical and Electrical Engineering, Jinling Institute of Technology, Nanjing 211169, China; 2College of Mechanical and Electrical Engineering, Wenzhou University, Wenzhou 325035, China

**Keywords:** stiffened curved panels, piezoelectric sensor array, calibrated 2D-MUSIC, impact localization

## Abstract

Stiffened structure-induced gain-phase errors degrade the performance of the high-resolution two-dimensional multiple signal classification (2D-MUSIC) algorithm, which makes it impossible to ensure the high accuracy of impact localization results. To eliminate the localization bias caused by these errors, a calibrated 2D-MUSIC-based impact localization method is first introduced. Firstly, time-frequency characteristics of the non-stationary impact signals are evaluated by experiment to obtain a clear first wave packet or a wave packet that purely corresponds to a single mode through continuous wavelet transform (CWT). Then, the uniform linear array covariance matrix with gain-phase errors is calibrated to be constructed as a Toeplitz structural matrix. By reconstructing covariance matrix R, 2D-MUSIC-based impact localization is calibrated for stiffened curved composite structures. Experimental research on the stiffened curved composite panel is carried out, and these impact localization results demonstrate the validity and effectiveness of the calibrated 2D-MUSIC-based method.

## 1. Introduction

An impact event on aircraft composite structures significantly reduces residual structural strength, especially compressive strength. Passive health monitoring is a technique to locate and characterize acoustic sources of material fractures caused by external impacts in real time [1]. Lamb wave-based methods can be effective alternatives due to their ability to select sensitive modes or frequencies, and their potential for online monitoring [2]. 

The sensor array-based MUSIC algorithm has been deeply researched and applied to the impact location of thin plates. Engholm [3] used one-dimensional far-field MUSIC to locate damage angles on metal structures using a uniform circular array and linear array. Yuan et al. [4] proposed a near-field 2D-MUSIC method to obtain the impact source angle and distance simultaneously for composite structures. The authors proposed a two-dimensional plum-blossom sensor array-based MUSIC method to realize omnidirectional impact localization for a simple composite plate [5]. As known to all, multimodal and dispersive characteristics of Lamb waves cause the wave packet to overlap in the time domain and frequency domain, which challenges the signal interpretation. Azuara et al. [6] pointed out that extracting meaningful physical features from the response signals is challenging due to the multimodal nature of Lamb waves and the geometric shape of the structure. Aircraft composite structures generally have stiffened components to enhance structural stiffness in practical applications. Stiffened-component scattering effects make received signals more complicated, which do not exist as a clear first wave packet or a wave packet purely corresponding to a single mode for analysis. Li et al. [7] proposed a method based on variational mode decomposition and wavelet transform to enhance and extract the location features of stator insulation damage signals of large motors. Hu et al. [8] investigated the wave propagation characteristics and analyzed the amplitude distribution curves and the directivity diagrams of A0/S0 modes for 30CrMo steel curved plates. Zheng et al. [1] numerically evaluated Lamb wave propagation characteristics to obtain mode conversion at the stiffener and delay after transmission through the stiffener. Zhu and Qing et al. [2] proposed a two-step calibration and monitoring method to locate acoustic sources for laminates with stiffeners based on a parameterized laminate model. However, for real stiffened composite structures, an accurate and timely method for impact localization is still challenging to realize.

Nevertheless, MUSIC algorithm-based methods are not guaranteed to yield optimal location on stiffened structures since they ignore the scattering effects caused by the stiffeners. The current literature usually compensated stiffened-component scattering effects with experimental investigations. Yuan et al. [9] reduced the localization error by an anisotropy compensated MUSIC algorithm on a stiffened composite panel. Ge et al. [10] introduced a gain-phase error calibration algorithm for the robust adaptive beamforming problem of linear arrays. However, direct experimental measurement campaigns in advance may be unfeasible due to their cost and the time required. Considering imprecise sensor array parameters, the authors constructed the cost function and proposed an adaptive piezoelectric sensor array self-calibration-based method for impact localization on a simple composite plate [11]. However, this self-calibration campaign needs an initial location to update the error matrix. Additionally, if the standard 2D-MUSIC algorithm is still adopted for stiffened plates, errors that cannot be neglected will be brought. 

Aiming at solving challenges in the signal interpretation of array signal, a piezoelectric sensor array-based impact location method through mode identification with continuous wavelet transform is introduced for a stiffened curved panel in this paper. Additionally, a calibrated 2D-MUSIC-based impact localization method is first introduced to eliminate the localization bias caused by stiffened structure-induced gain-phase errors. This paper is organized as follows: Section 2 introduces the process of the calibrated MUSIC algorithm on the stiffened structures for impact localization. In Section 3, the impact localization experiment is performed on a complex stiffened curved panel and the details of impact localization are depicted and the mode identification and feature extraction with CWT are proposed. Eventually, Section 4 gives the conclusion. 

## 2. Calibrated Sensor Array-Based Localization Method

The guided waves, induced by acoustic emission-like impact, propagating on curved panels which have a radius of curvature much larger than the wall thickness are very similar to Lamb waves. It is expected that the array sensor-based imaging method could work sufficiently well on curved panels.

### 2.1. Array Sensor Signal Model

As seen in Figure 1, a curved panel has a radius of curvature R, chord length *L*, and arc length *L*. The observed signal consists of a uniform linear array of 2M+1 piezoelectric sensors with a space of *d*. Assuming the narrowband signal $x_{q}(t)$ induced by acoustic emission source s(t) with a certain frequency $\omega_{0}$ of PZT Sq under the near-field situation can be presented as
(1)$$x_{q}(t)=\frac{L^{a}}{L_{q}^{a}}s(t)e^{-j\omega_{0}\tau_{q}}+n_{q}(t),\;q=-M,\cdots ,0,\cdots ,M$$
where $n_{q}(t)$ express the background noise, $L^{a}$ and $L_{q}^{a}$ are the arc length between acoustic emission source, PZT S0 and PZT Sq, which can be calculated with the chord using the formula,
(2)$$L_{q}^{c}=2R\sin\frac{L_{q}^{a}}{2R}$$

Additionally, the chord length $L_{q}^{c}$ between the impact source and PZT Sq is calculated as
(3)$$L_{q}^{c}=\sqrt{{\left(L^{c}\right)}^{2}+d^{2}{\left(q-1\right)}^{2}-2rd(q-1)\cos\theta }$$

The impact signal arriving time difference between PZT Sq and PZT0 is defined as
(4)$$\tau_{q}=(L^{a}-L_{q}^{a})/c$$
where c is the average velocity. The steering vector of PZT Sq is denoted as
(5)$$a_{q}(L^{c},\theta )=\frac{L^{a}}{L_{q}^{a}}\exp(j\omega_{0}\tau_{q})$$

For the whole array, the response signals can be presented as
(6)X(t)=A(r,θ)X(t)+N(t)
where
$$\begin{array}{l} X(t)={\left[x_{-M}(t),\cdots ,x_{q}(t),\cdots ,x_{M}(t)\right]}^{T}\\ A(r,\theta )={\left[a_{-M}(r,\theta ),\cdots ,a_{q}(r,\theta ),\cdots ,a_{M}(r,\theta )\right]}^{T}\\ N(t)={\left[n_{-M}(t),\cdots ,n_{q}(t),\cdots ,n_{M}(t)\right]}^{T} \end{array}$$

### 2.2. Calibrated Sensor Array-Based Localization

The covariance matrix of the observed signal vector from the sensor array is then decomposed to obtain the signal subspace and noise subspace, which using the ideal covariance matrix of the whole array response signals is
(7)$$R_{XX}=E[X(t)X^{H}(t)]=AR_{SS}A^{H}+\delta^{2}I$$

In Formula (7) *H* is the conjugate transpose, $R_{ss}=\mathrm{diag}[r_{1}(t),r_{2}(t),\cdots ,r_{N}(t)]$ can be approximated as a diagonal matrix, $\delta^{2}$ means the power of the signal source, and $AR_{SS}A^{H}$ expresses the noise power. For a uniform linear array, the elements of the *m*^th^ row and *n*^th^ column of the covariance matrix $R_{XX}$ can be represented as
(8)$$R_{XX}(m,n)=\sum_{q=1}^{N}L_{q}^{c}\exp(j2\pi (m-n)d\sin\theta_{q}/c)+{\delta^{2}}_{mn}$$
when the signal-to-noise ratio is large, the influence of noise on the covariance matrix can be ignored. Thus, it can be known that the covariance matrix is a Hermite Toeplitz matrix under the condition of an ideal uniform linear array model. 

When there are amplitude and phase errors Γ induced by a stiffened structure, the data covariance is expressed as
(9)$${\hat{R}}_{XX}=E[X(t)X^{H}(t)]=\Gamma AR_{SS}A^{H}\Gamma^{H}+\delta^{2}I$$

The elements of the mth row and nth column of the covariance matrix ${\hat{R}}_{XX}$ can be written as
(10)$${\hat{R}}_{XX}(m,n)=\sum_{i=1}^{N}\Gamma_{m}\Gamma_{n}L_{q}^{c}\exp(j2\pi (m-n)d\sin\theta_{q}/c)+{\delta^{2}}_{mn}$$

It can be seen from the above formula that when the amplitude and phase errors exist, the covariance matrix no longer satisfies the structural characteristics of the Hermite Toeplitz matrix. According to the structural characteristics of the covariance matrix in the case of no error, the covariance matrix with amplitude and phase errors is normalized, which means that the average value of the elements in the matrix parallel to the main diagonal is calculated to replace the original value. A new covariance matrix ${\hat{R}}_{TT}$ is obtained after the matrix transformation, in which the elements of the mth row and the nth column of the matrix can be expressed as
(11)$${\hat{R}}_{TT}(m,n)=\frac{\sum_{m,n=1}^{2M+1}{\hat{R}}_{XX}(m,n)}{2M+1-K},\left|n-m\right|=K,k=0,\cdots ,2M$$

The transformed covariance matrix is decomposed into eigenvalues, and the signal subspace is $U_{S}=[{\hat{e}}_{1},{\hat{e}}_{2},\cdots {\hat{e}}_{N}]$ and the noise subspace is $U_{N}=[{\hat{e}}_{N+1}\cdots {\hat{e}}_{M}]$.

The orthogonality of two subspaces can be utilized to estimate the signal parameters. To describe the orthogonal properties, the spatial spectrum is used, which can be calculated by
(12)$$P_{\mathrm{MUSIC}}=\frac{1}{A^{H}(L^{c},\theta )U_{N}U_{N}{}^{H}A(L^{c},\theta )}$$

We vary $\left(L^{c},\theta \right)$ to realize a scanning process. When the steering vector matches the real source signal vectors, the denominator of Equation (12) approaches zero due to the orthogonal properties, resulting in a peak in the spatial spectrum $P_{\mathrm{MUSIC}}$.

## 3. Experiments on a Stiffened Curved Composite Panel

### 3.1. Experiment Setup

The experiment is conducted on a stiffened curved composite panel with 2 mm thickness, 500 mm length, and 475 mm width. The composite panel is made of T300 carbon fiber with five stiffeners evenly arranged along the curved edge, as shown in Figure 2. The array used in the experiment is a uniform linear sensor array bonded on the structure surface with seven piezoelectric (PZT) sensors. The PZT sensors’ diameter and thickness are 8 mm and 0.48 mm respectively. They are arranged with a space of 10 mm to meet the condition d≤λ/2, and are labeled as PZT S1 to PZT S7 respectively from the left to the right. The integrated structural health monitoring scanning system (ISHMS) is applied to sense acoustic emission signals. Low-velocity impacts induced by an impact hammer are performed at selected positions on this structure, as shown in Figure 2. The sampling rate is set to 2 MHz and the sampling length to 10,000 including 2000 pre-trigger samples.

### 3.2. Typical Lamb Wave Signal Analysis

In this section, typical impact response signals are chosen for time analysis and time-frequency analysis. When this impact occurs and propagates through the first to fifth stiffener, the typical outputs in the time domain of the PZT S4 sensor are shown in Figure 3. It shows that the received signals are complicated and have more mode conversions and dispersive behavior when crossing the second T-stiffeners than the first, and this phenomenon becomes more and more serious when the wave crosses more T-stiffeners.

According to the sensor array-based method, the signal model requires that the input signals should be narrow-band frequencies. However, the waves caused by low-velocity impact are wide-band signals. CWT can modify the length of the wavelets at different scales to adaptively analyze signals. The Gabor function was adopted in this study because it is known to provide a better resolution both in the time and frequency domain than any other wavelets [12]. Here, the Gabor wavelet transform is applied to analyze the non-stationary impact signals of this stiffened curved composite panel, and the time-frequency spectrogram is obtained using CWT, as seen in Figure 4. It shows that a wave packet appears between 4 kHz and 5 kHz in the first time-frequency spectrogram, but there cannot exist a clear first wave packet in the other four spectrograms due to the presence of the dispersive wave below 4 kHz. In order to reduce the influence of low-frequency signals, the time-frequency spectrogram after filtering is obtained in Figure 5. A wave packet obviously exists in the first four spectrograms, but it is still not clear in the last one. For the outputs of the PZT S4 when the impact signal crosses the fifth T-stiffener, the time-frequency spectrogram after further filtering is obtained in Figure 6, and a wave packet appears around 8 kHz. Using continuous wavelet transform, we can extract a narrowband signal that purely corresponds to the frequency and mode of interest for analysis.

### 3.3. Gain-Phase Errors of Sensor Array Signal

The impact at the point when it across 2nd T-stiffener as a typical case to be analyzed. Figure 7 shows the time-frequency spectrum of PZT S4 which is a typical case that represents the frequency characteristics of the impact-induced elastic signal obtained by the PZT sensors. The wave fronts of the A0 and S0 mode of the signals can be decided by checking the velocities of these two modes when excited [13]. Figure 8b shows the filtered 4.1 kHz frequency of the typical impact case, and the direct wavefronts that purely correspond to the A0 mode can be found between 0.1 ms and 0.3 ms, which can be extracted for analysis and used as the input vector of the 2D-MUSIC model.

To investigate the gain-phase errors’ effects on Lamb waves, the Gabor wavelet was chosen to extract the wavelet envelope curves at a specific frequency. The array responded signals were obtained by CWT filtering of the 4.1 kHz frequency component at the simulated impact source position shown in Figure 8. The peak of the direct wave is selected to measure the gain-phase difference. Similarly, we can obtain the gain-phase errors when Lamb waves cross different numbers of T-stiffeners obstacles, as demonstrated in Figure 9.

## 4. Conclusions and Future Works

According to Figure 2, ten impacts at different points are investigated whose numbers and actual positions are listed in Table 1. The direct wavefronts that correspond to the A0 mode are extracted and used as the input vector of the 2D-MUSIC model. According to the structure dimension, the area scanned is set to be the distance from 0 to 450 mm and the direction from 0° to 180°. A spatial spectrum $P_{MUSIC}(L^{c},\theta )$ obtained is shown in Figure 10. In the figure, the color represents the spatial spectrum magnitudes of each scanned point $\left(L^{c},\theta \right)$. The deepest point represents the impact point localized by the calibrated MUSIC algorithm. The whole impact localization process cost 5~6 s under the condition of dual CPU 1.46°GHz and 1G memory card. 

According to the continuous wavelet transform on typical outputs of the PZT sensors after filtering, the suitable frequencies of 4.1 kHz, 4.8 kHz, 5.0 kHz, and 8 kHz of typical outputs of the PZT sensors are selected when impact signals cross one, two, three, and five T-stiffeners, respectively. Those purely wave packets correspond to A0 mode and are substituted into its steering vector to estimate the azimuth of the signal source. Four impacts, including (a) No. 3, crossing one stiffener, (b) No. 13, crossing two stiffeners, (c) No. 23, crossing three stiffeners, and (d) No. 38, crossing five stiffeners, are shown in Figure 10, which are in good agreement with the actual impacts. The maximum error is at the No. 38 impact position where Lamb waves cross five stiffeners. Its direction and distance error is 1° and 2.4 cm respectively.

Ten low-velocity impacts are performed at various positions marked with numbers on the structure shown in Figure 2. To guarantee the time resolution and narrowband signal extraction, a suitable frequency and time are firstly extracted as the central frequency in the narrowband signal extraction processing using CWT. Additionally, they are substituted into the steering vector to estimate the impact positions. The location results in Polar coordinates using the 2D-MUSIC algorithm which propagates through different directions of stiffened parts and their Cartesian coordinates are listed in Table 1. All estimated results errors are compared with actual impact points and are also listed in Table 1. For impact localization results obtained by the proposed method, the maximum error of distance estimation is less than 2.4 cm and the maximum error of direction estimation is less than 3°. In Cartesian coordinates, the maximum error of x is less than 1.3 cm and the maximum error of y is less than 2.4 cm. Selected impact locations results in Cartesian coordinates are graphically shown in Figure 11. The experiment results verify the piezoelectric sensor array-based impact location method through mode identification with the proposed continuous wavelet transform can effectively monitor stiffened curved composite panels with high accuracy.

A piezoelectric sensor array-based impact location method through mode identification with CWT is introduced for a stiffened curved panel. Firstly, a time-frequency spectrogram is obtained using CWT of Gabor wavelet transform to analyze the impact signals for a stiffened curved composite panel and to extract a narrowband signal that purely corresponds to the frequency and mode of interest for analysis. Secondly, those purely wave packet direct wavefronts corresponding to the A0 mode are substituted into the steering vector of the 2D-MUSIC model to estimate the impact signal source. The validity and effectiveness of the proposed method are experimentally demonstrated on a stiffened curved panel. Five impacts in which Lamb wave induced by impact across one stiffener, two stiffeners, three stiffeners, four stiffeners, and five stiffeners, are investigated for stiffener effect. Then, ten low-velocity impacts are performed at various positions marked with numbers on the structure and have good agreement with the actual impacts. The maximum error of distance estimation is less than 2.4 cm and the maximum error of direction estimation is less than 3°. In Cartesian coordinates, the maximum error of x is less than 1.3 cm and the maximum error of y is less than 2.4 cm. The experiment results verify the piezoelectric sensor array-based impact location method through mode identification with continuous wavelet transform can effectively monitor stiffened curved composite panels with high accuracy. Further research is still worth doing to systematically address the adaptive mode identification and feature extraction to shorten the time of impact localization on complex structures.

## Figures and Tables

**Figure 1 sensors-22-05879-f001:**
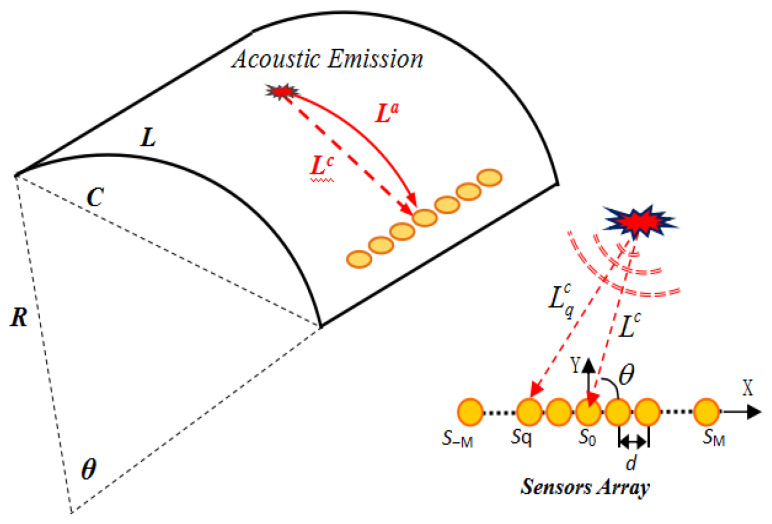
Acoustic emission signal model of a curved panel.

**Figure 2 sensors-22-05879-f002:**
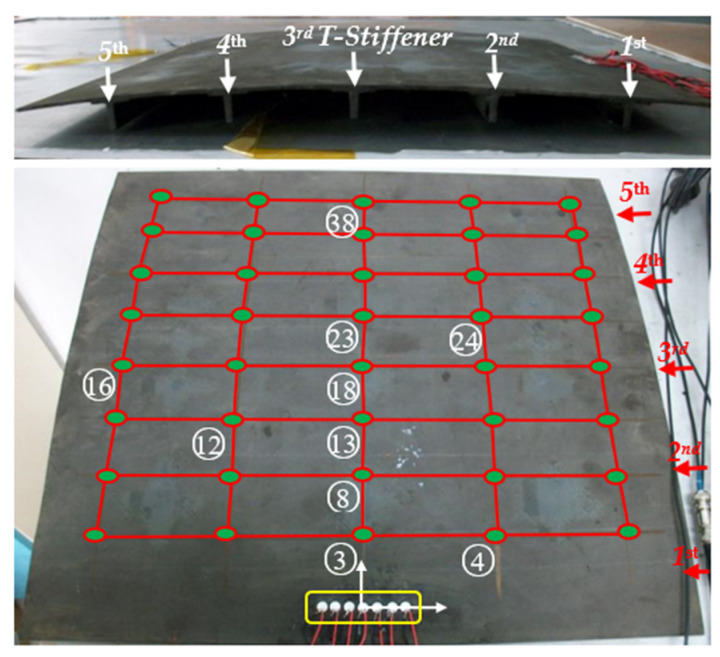
Impact location experiment on a curved composite panel with five stiffeners.

**Figure 3 sensors-22-05879-f003:**
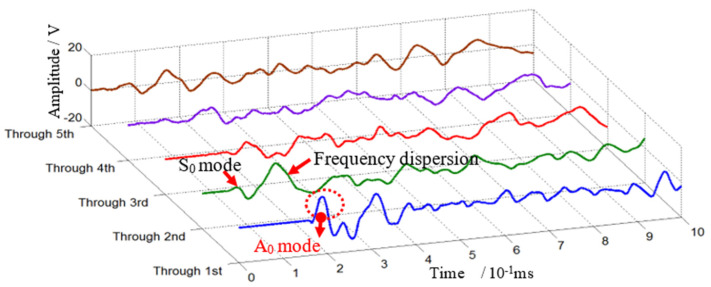
Typical outputs of the PZT sensors when impact signals cross these T-stiffeners.

**Figure 4 sensors-22-05879-f004:**
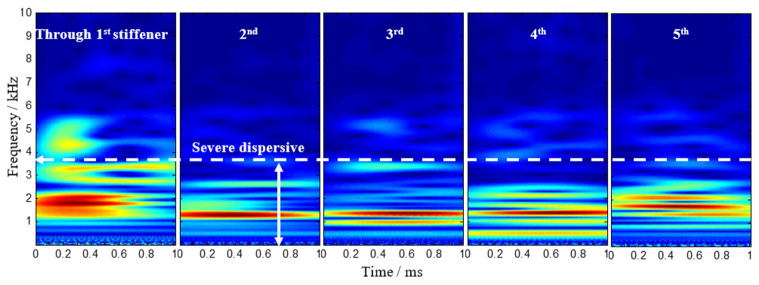
Continuous wavelet transforms on typical outputs of the PZT sensors.

**Figure 5 sensors-22-05879-f005:**
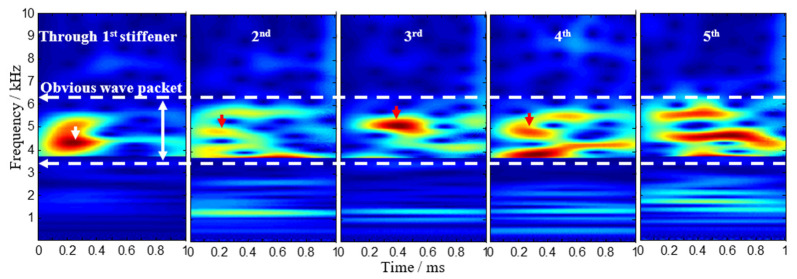
Continuous wavelet transforms on typical outputs of the PZT sensors after filtering.

**Figure 6 sensors-22-05879-f006:**
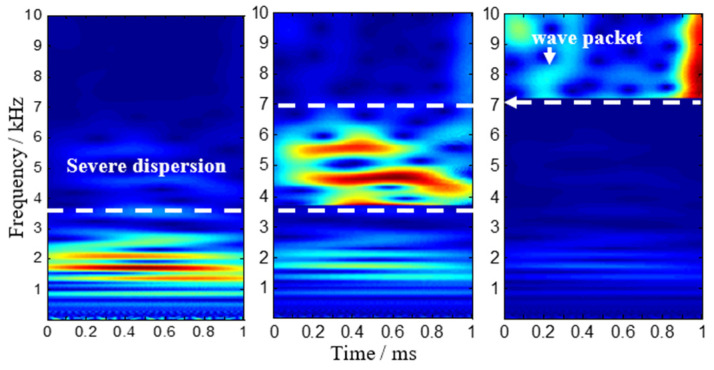
Continuous wavelet transforms on outputs of the PZT S4 when crossing 5th T-stiffeners.

**Figure 7 sensors-22-05879-f007:**
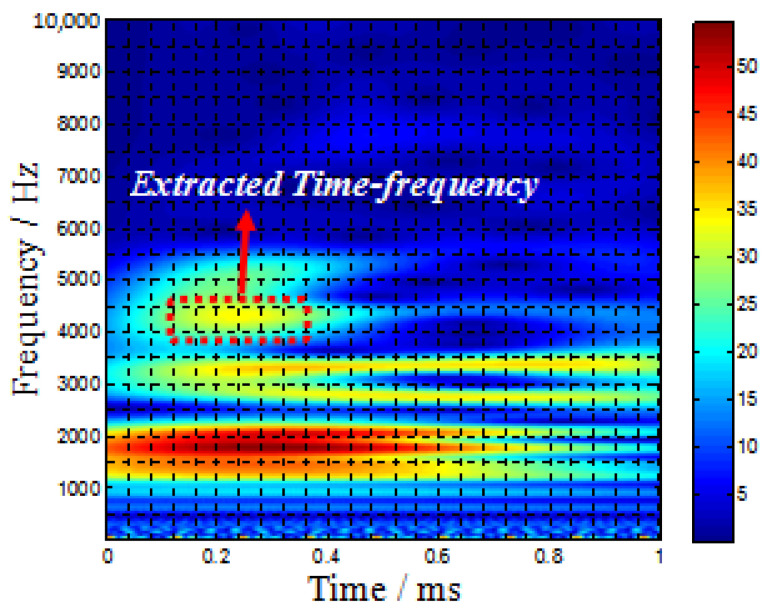
Continuous wavelet transform analysis when impact happens at No. 3 position.

**Figure 8 sensors-22-05879-f008:**
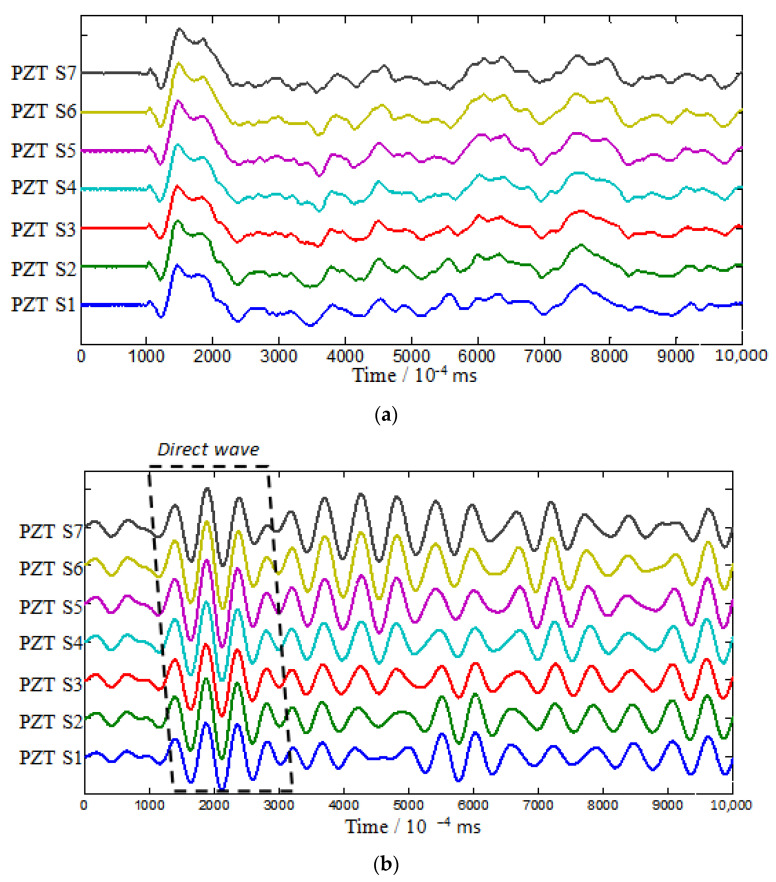
(**a**) Raw signal without filtration; (**b**) CWT filtering of the 4.1 kHz frequency component.

**Figure 9 sensors-22-05879-f009:**
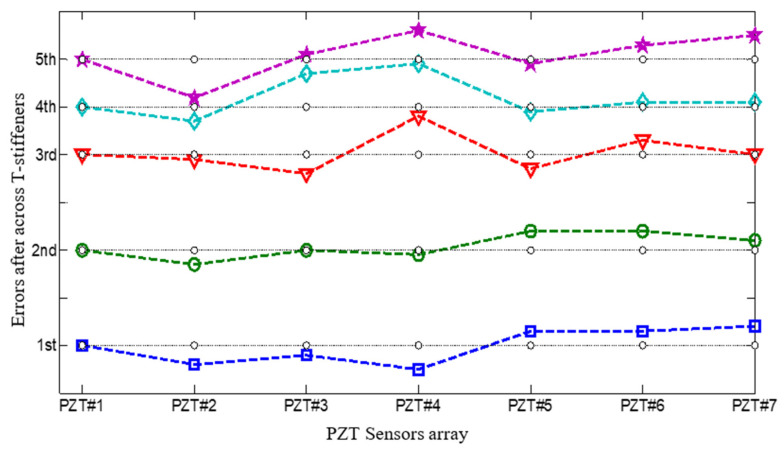
Phase errors when Lamb waves cross T-stiffeners.

**Figure 10 sensors-22-05879-f010:**
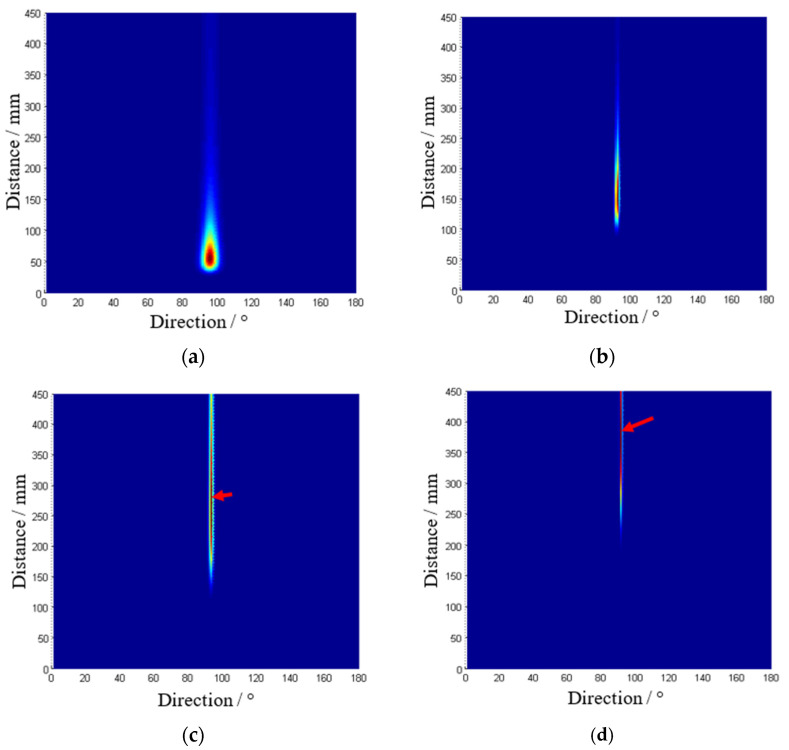
The spatial spectrums estimated by the 2D-MUSIC algorithm. (**a**) No. 3; (**b**) No. 13; (**c**) No. 23; (**d**) No. 38.

**Figure 11 sensors-22-05879-f011:**
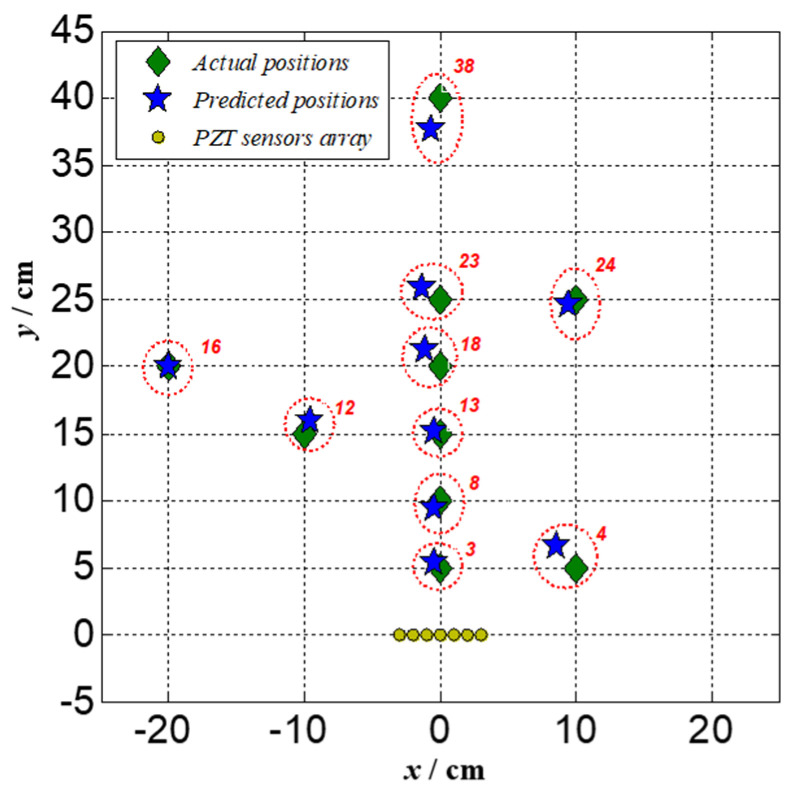
Impact location graphical results in Cartesian coordinates.

**Table 1 sensors-22-05879-t001:** Feature extracted and impact localization results on the curved composite panel.

No.	Actual Positions		Feature Extracted		Predicted Positions and Errors					
	x/cm	y/cm	f/kHz	t/ms	$\hat{r}$/mm	$\hat{\theta }$/°	$\hat{x}$/cm	$\hat{y}$/cm	$E_{x}$/cm	$E_{y}$/cm
3	0	5	4.1	1.3~1.685	107	93	−0.4	5.4	0.4	0.4
4	10	5	4.8	1.515~1.7	122	29	8.6	6.6	1.4	1.6
8	0	10	6.0	1.08~1.34	110	93	−0.5	9.4	0.5	0.6
12	−10	15	6.5	0.7~2.04	187	121	−9.6	16	0.4	1
13	0	15	4.8	0.6~1.225	161	92	−0.5	15.2	0.5	0.2
16	−20	20	4.2	0.81~1	283	135	−20	20	0	0
18	0	20	6.0	0.45~1.05	212	93	−1.1	21.2	1.1	1.2
23	0	25	5.0	0.45~0.93	259	93	−1.3	25.8	1.3	0.8
24	10	25	4.0	1.11~1.3	264	69	9.5	24.6	0.5	0.4
38	0	40	8.0	0.1~0.65	376	91	−0.7	37.6	0.7	2.4

## Data Availability

The data that support the findings of this study are available from the corresponding author upon reasonable request.

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
