# Peer review of "Gain-Phase Error-Calibrated Piezoelectric Sensor Array-Based Impact Localization on Stiffened Curved Composite Structures"

_sensors, 2022, doi:10.3390/s22155879_

Round 1

Reviewer 1 Report

Reviewed paper is related to impact location in curved stiffened composite panel. Authors utilise MUSIC algorithm and wavelet filtration method. Bellow I have listed my comments:

It is hard to determined what is really new in this research.  I understood that filtering of broadband impact signals?

Title very long, please remove „Research on” and try to make it shorter

“propagating on curved 135 panels which has a radius of curvature much larger than the wall thickness are very similar to the Lamb waves”  What kind of influence panel curvature has on elastic wave propagation?  Please compare case of flat panel with curved (I mean discussion).

Sensors details: thickness, material type

Data of panel material: stacking sequence, number of layers,…

Why authors use 10 mm transducer spacing? Is there any correlation to wavelength of utilised wave mode? What is the influence of spacing on results of impact localisation?

Why authors utilised 7 transducers not for example 10? What is the influence of added/removed sensors? Is there optimal number of transducers?

Fig. 2 – How authors determined that there are A0 and S0 modes? What kind of analysis did authors perform?  These signals have very small SNR and looks like noises

Conclusions: “piezoelectric sensors array based impact location method through mode identification with CWT is introduced for a stiffened curved panels” The same question, what kind of identification was conducted? Did authors analysed dispersion curves?

Page 5: “and the direct 120 wave fronts purely corresponds to the A0 mode can be found”. How did author determined that it is A0 mode?

Fig 7 Shows filtered signal but please show also raw signal (without filtration). Why Gabor wavelet? Why authors utilised wavelet-based filtering? Did authors try to use simple digital filters (high pass)?

Energy of impact induced guided waves is located in low frequency band (few kHz). How this situation look in other research papers, related to similar composite structures with similar thickness?

Please carefully review English:

“When this impact occurs and propagating though 1st ~5th stiffener…” – impact is propagating of wave induced by impact?

“Though” – should be “through”

Reviewer 2 Report

Overview:

The authors present a method for determining the location of an impact on a composite panel with stiffeners. The method relies on wavelet transform to filter out the wave mode of interest from the wideband signals that are usually result in the event of impact. The filtered signals are then fed into the 2D-MUSIC algorithm to determine the azimuth and the distance to the impact location. The authors present experimental application of the proposed technique with several scenarios with different impact locations with respect to the number of stiffeners located between the impact location and the positions of sensors.

Overall, the reviewer believes that the manuscript presents a valuable advancement in the field of structural health monitoring and recommends the manuscript is published after addressing the following queries.

Major queries:

1.       Please clearly explain how this work is different from the previously published work by two of the coauthors - Ren, L.; Zhong, Y.; Xiang, J.; Wang, Z. Adaptive Sensor Array Error Calibration Based Impact Localization on Composite Structure. Appl. Sci. 2020, 10, 4042. https://doi.org/10.3390/app10114042

Somehow, this is not included in the introduction and literature review. Please expand the literature review to include relevant work in it.

2.       Define “r” in Equation (3). Do you mean “R”?

3.       Define capital gamma in Equation (9).

4.       Conclusions are presented in Section 4.

5.       The reviewer suggests to rearrange the order of the sections. The algorithm presented in Section 3 should go before the description of experiments in Section 2.

6.       Please add a discussion of computational requirements for performing impact localization. Is it possible to do in real time on an aircraft? What kind of hardware will it require? Etc.

Minor queries:

1.       Please correct “Localization” – should be lower case, line 19.

2.       Line 143 – check reference to figure 1, which should probably be fig. 9.

3.       Line 151 – check capitalization.

4.       Line 199 – check reference to Figure 2.

5.       The reviewer suggests that the authors perform a thorough revision of the manuscript using a professional English editing service. In the current form the manuscript is difficult to read due to many grammatical and other issues with use of English language.

Round 2

Reviewer 1 Report

Dear Authors thank you very much for answer to all my questions and paper improvement. Now, this paper could be published in the Sensors Journal. Good luck!

Reviewer 2 Report

Thank you for addressing the reviewer's comments and improving the paper. The paper still needs additional proofreading to improve English language and readability.

For example: page 1, line 42: "It is difficult to extract the features of Lamb wave signals which significantly reducing the accuracy of the damage located [6]." This sentence is difficult to understand and needs revision.

Line 44: "... there is a challenging for extracting meaningful physical features from the response signals due to the multi-modal nature of Lamb wave, the geometric shape of the structure and their environmental dependency." - Again, this needs revision.

There are many other examples through the paper.

There are also minor issues, for example:

- missing spaces between values and units "500mm length and 475mm", on lines 138-139.

- Line 210: "... are shown in Figures which are in good..." - missing reference to Figure 11?
